# Effect of minimally invasive autopsy and ethnic background on acceptance of clinical postmortem investigation in adults

I. M. Wagensveld[1,2]*, A. C. Weustink[1,2], J. A. Kors[3], B. M. Blokker[1,2], M. G. M. Hunink[1,4,5], J. W. Oosterhuis[2]

1 Department of Radiology and Nuclear Medicine, Erasmus University Medical Centre, Rotterdam, The Netherlands, 2 Department of Pathology, Erasmus University Medical Centre, Rotterdam, The Netherlands, 3 Department of Medical Informatics, Erasmus University Medical Centre, Rotterdam, The Netherlands, 4 Department of Epidemiology and Biostatistics, Erasmus University Medical Centre, Rotterdam, The Netherlands, 5 Centre for Health Decision Science, Harvard T.H. Chan School of Public Health, Harvard University, Boston, Massachusetts, United States of America

* i.wagensveld@erasmusmc.nl

**Data Availability Statement:** Some data cannot be shared publicly because of the confidential nature (i.e. surnames). The IRB/ Ethics committee

## Abstract

### Objectives

Autopsy rates worldwide have dropped significantly over the last five decades. Imaging based autopsies are increasingly used as alternatives to conventional autopsy (CA). The aim of this study was to investigate the effect of the introduction of minimally invasive autopsy, consisting of CT, MRI and tissue biopsies on the overall autopsy rate (of CA and minimally invasive autopsy) and the autopsy rate among different ethnicities.

### Methods

We performed a prospective single center before-after study. The intervention was the introduction of minimally invasive autopsy as an alternative to CA. Minimally invasive autopsy consisted of MRI, CT, and CT-guided tissue biopsies. Autopsy rates over time and the effect of introducing minimally invasive autopsy were analyzed with a linear regression model. We performed a subgroup analysis comparing the autopsy rates of two groups: a group of western-European ethnicity versus a group of other ethnicities.

### Results

Autopsy rates declined from 14.0% in 2010 to 8.3% in 2019. The linear regression model showed a significant effect of both time and availability of minimally invasive autopsy on the overall autopsy rate. The predicted autopsy rate in the model started at 15.1% in 2010 and dropped approximately 0.1% per month (β = -0.001, p < 0.001). Availability of minimally invasive autopsy increased the overall autopsy rate by 2.4% (β = 0.024, p < 0.001). The overall autopsy rate of people with an ethnic background other than western-European was significantly higher in years when minimally invasive autopsy was available compared to when it was not (22/176 = 12.5% vs. 81/1014 (8.0%), p = 0.049).

imposing data sharing restrictions is The Medical Ethics Review Committee (MERC).These data are available upon request at the Erasmus Medical Center Clinical trial bureau of the radiology department (contact via imaging. trialbureau@erasmusmc.nl) for researchers who meet the criteria for access to confidential data

**Funding:** This work was supported by Erasmus Medical Centre Health Care Efficiency (grant 2010-10112), Erasmus MC Vriendenfonds (grant 104117), and Stichting Coolsingel (grant 255). The funders had no role in study design, data collection and analysis, decision to publish, or preparation of the manuscript.

**Competing interests:** The authors have declared that no competing interests exist.

## Conclusions

The introduction of the minimally invasive autopsy had a small, but significant effect on the overall autopsy rate. Furthermore, the minimally invasive autopsy appears to be more acceptable than CA among people with an ethnicity other than western-European.

## Introduction

Clinical autopsy rates have dropped considerably over the previous decades. [1–6] Improvements in imaging techniques in living patients may have led to the belief that autopsies hardly add to the information acquired prior to death. [7] However, despite improved diagnostics, autopsies still provide valuable feedback on diagnoses and treatment, and accurate statistics on causes of death. [8] Moreover, they are useful for healthcare policymaking, education and research purposes. [3, 9–11]

The low consent rate of next-of-kin for the autopsy is one of the reasons for the decline in clinical autopsy rates. Therefore, strategies to improve the consent rates are under investigation: improved availability of modern imaging techniques has led to the development of imaging-based autopsy techniques. Such methods can be non-invasive or minimally invasive [12]; in our hospital we introduced and validated a minimally invasive autopsy, consisting of postmortem CT, MRI and CT-guided biopsies. [8]

The aim of the present study was to determine the overall autopsy rate (minimally invasive autopsy and conventional autopsy) on adult deceased patients in our hospital over the years 2010–2019 and to investigate whether the introduction of the minimally invasive autopsy led to an increase in overall autopsy rate. A subgroup analysis was performed comparing a group of western-European ethnicity with a group of other ethnicities. Furthermore, we used a questionnaire to investigate the motivations of doctors and next-of-kin in the consent process for postmortem investigation.

## Materials and methods

### Setting and design

This study was performed at the Erasmus University Medical Center in Rotterdam, the largest academic hospital in the Netherlands. The design was a prospective before-after study whereby additional data was collected retrospectively. The intervention was the introduction of the minimally invasive autopsy as an alternative to conventional autopsy. The study was approved by the Erasmus University Medical Center Medical Ethical Committee (file number MEC-2011-055). The institutional review board approved the study prior to data collection. All adult patients who had died in-hospital were included.

### Autopsy procedures

Conventional autopsy and minimally invasive autopsy were both available from Monday to Friday. Only 1 minimally invasive autopsy per day was possible, due to limited scanner availability. When multiple minimally invasive autopsy procedures were requested on the same day, requests would be processed in the order they were received: if next-of-kin agreed, a procedure would be postponed until the next available day.

## Minimally invasive autopsy procedure

A minimally invasive autopsy consisted of MRI of the head and torso, full-body CT scan and CT-guided biopsies of organs (heart, lungs, liver, kidneys and spleen) and additional biopsies of abnormal / pathological lesions detected on imaging. The main difference between the conventional autopsy and the minimally invasive autopsy is that minimally invasive autopsy leaves the body intact, whereas conventional autopsy is an invasive procedure where the body is opened with a Y-incision and the organs are subsequently removed and dissected. More details about the minimally invasive autopsy procedure, including CT and MRI protocols are described in previous articles. [8, 13–15]

## Conventional autopsy procedure

The conventional autopsy was performed according to standard department protocol: the body was opened with a Y-incision and the thoracic cavity opened with a rib-cutter. Organs were eviscerated by the mortuary assistant and dissected by a resident in pathology, supervised by a certified pathologist. [8]

## Primary outcome

The primary outcome measure was the effect of minimally invasive autopsy on the overall autopsy rate, the term we will use from here on to describe the combined autopsy rates of conventional autopsy and minimally invasive autopsy.

Consent for both autopsy procedures was requested by the treating physicians. Before the actual introduction of the minimally invasive autopsy, all clinical wards were educated about the new autopsy method.

## Secondary outcome

As a secondary outcome we investigated the motivations of next-of-kin for consenting to or refusing an autopsy, and of doctors to not ask for permission. We distributed the questionnaires to the doctors who were involved in the consent process, since they would be informed about the motivations of next-of-kin when discussing the possibility of autopsy after a patient had passed away. Questionnaires were distributed from September 2016—December 2017.

## Data analysis

Because we expected that the autopsy rates would decline during the study period we performed a linear regression analysis to calculate the effect of time and availability of minimally invasive autopsy on overall autopsy rate. The independent variables were time in months since the start of the study and the availability of minimally invasive autopsy as a standalone post-mortem investigation.

We performed a subgroup analysis among people of western-European ethnicity versus people of other ethnical backgrounds. We calculated the overall autopsy rates of both groups and performed an independent T-test in order to test for significance.

To determine the ethnicity, we used a two-stage classification process. In the first stage, the predicted probabilities of a supervised machine-learning algorithm were used to distinguish between classifications with high and low certainty. For this stage, we used a random forest classifier (method ranger in the R package caret). To train and test the classifier, we used data from a questionnaire in which the next-of-kin were asked to provide the ethnic background of the deceased. [7, 8] The total set consisted of 2,083 cases, which were split in a training set (80%) and a test set (20%). As features we used character n-grams (with n = 2, 3, and 4) of the

last names and, if available, of the first names. Each case was labelled as having either a western-European or other ethnic background. Henceforth we will refer to this latter group as 'other' ethnicities. The training set was used to develop the classifier; the test set was only used for performance evaluation. Performance measures were the area under the receiver operating characteristic curve (AUROC), negative predictive value (proportion of correctly predicted western-European cases), and positive predictive value (proportion of correctly predicted cases of 'other' ethnicity). [16–18] In the second stage, the classifications with low certainty were manually validated, in cases of doubt the Dutch surname database was consulted (https://www.cbgfamilienamen.nl/nfb/). For the manual classification, the group allocation (intervention vs. non-intervention) was unknown to the observer (IMW).

## Results

### Acceptance

Autopsy rates declined from 14.0% in 2010 to 8.3% in 2019. The annual autopsy rates of the years 2010–2019 are shown in Fig 1.

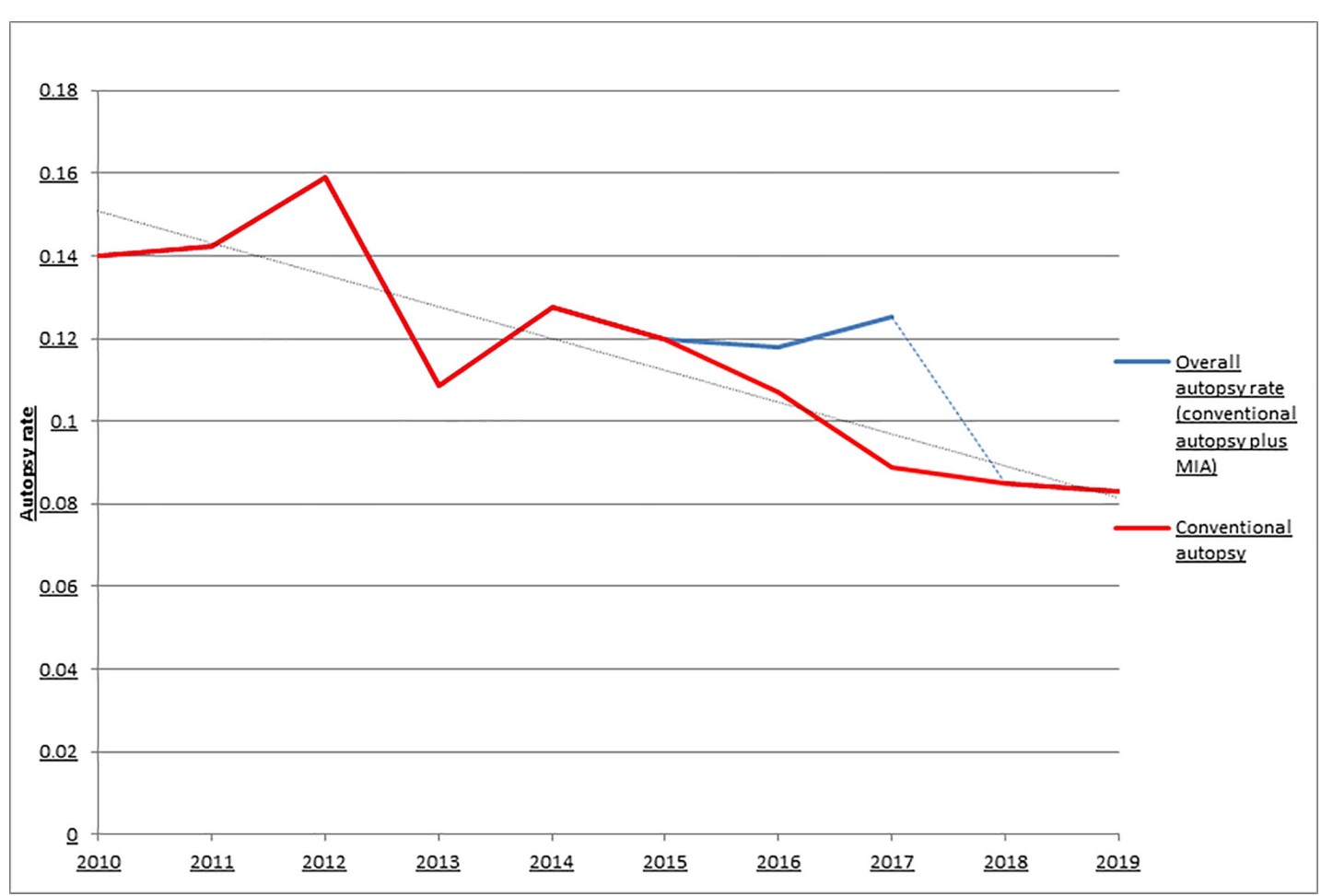

**Fig 1. Autopsy rates 2010–2019.** The overall autopsy rate consists of the combination of the autopsy rates of conventional autopsy and minimally invasive autopsy. The blue dashed line represents the return of the autopsy rate to the trendline after the minimally invasive autopsy was no longer available in our hospital.

In the 'intervention' period (when minimally invasive autopsy was available) from October 2016 through December 2017, 1056 adult patients died and permission for postmortem diagnostics was given in 133 cases (12.6%): 87 underwent conventional autopsy (8.2%) and 46 minimally invasive autopsy (4.4%).

The linear regression model showed a significant effect on the overall autopsy rate of both time and availability of minimally invasive autopsy. The predicted autopsy rate in the model started at 15.1% in September 2010 and dropped approximately 0.1% per month ($\beta$ = -0.001, p < 0.001) and minimally invasive autopsy availability increased the overall autopsy rate by 2.4% ($\beta$ = 0.024, p < 0.001).

## Prediction of ethnicity

We used 5-fold cross-validation to train the random forest classifier on the training set. On the test set, this classifier obtained an area under the receiver operating characteristic curve of 0.91, showing good performance. We empirically set probability thresholds to distinguish between high and low confidence classifications. Names with a predicted probability lower than 0.50 were labeled as 'other' ethnicity (this threshold yielded a positive predictive value of 1 on the test set), and names with a predicted probability greater than 0.82 were labeled as western-European (yielding a negative predictive value of 0.95 on the test set). Cases with a probability between 0.50 and 0.82 were manually validated, using the Dutch surnames database. Of the cases from 2010–2019 which were not part of the learning dataset (n = 4764), 1548 cases had a probability between 0.5 and 0.82; 733 (47.4%) of these were manually scored as western-European. In the total cohort, 82.8% (5736/6928) were classified as western-European.

## Effect of ethnic background on acceptance

The autopsy rates of western-Europeans and 'other' ethnicities in the different cohorts is detailed in Table 1. In the years when minimally invasive autopsy was not available, western-Europeans had a significantly higher autopsy rate than 'other' ethnicities (629/4858 = 12.9% vs. 82/1014 = 8.0%, p<0.001). When minimally invasive autopsy was available the overall autopsy rate was nearly the same for western-Europeans and 'other' ethnicities (111/880 = 12.6% vs. 22/176 = 12.5%, p = 0.97). The overall autopsy rate among 'other' ethnicities was significantly higher in the years when minimally invasive autopsy was available compared to the years when it was not (22/176 = 12.5% vs. 81/1014 = 8.0%, p = 0.049).

**Table 1. Autopsy rates of western-European vs 'other' ethnicities.**

| Time period | Available procedures | n | | western-European | 'other' ethnicities | Overall autopsy rate | western-European vs 'other' ethnicities |
|---|---|---|---|---|---|---|---|
| Pre-intervention (Jan. 2010—Sept. 2016) | CA only | 4679 | CA rate | 538/3875 (13.9%) | 71/804 (8.8%) | 609/4679 (13.0%) | p>0.001 |
| Post-intervention (Jan. 2018- Mar. 2019) | CA only | 1193 | CA rate | 90/982 (9.2%) | 11/211 (5.2%) | 101/1193 (8.5%) | p = 0.061 |
| Total non-intervention (2010–2019, excluding intervention period) | CA only | 5872 | CA rate | **629/4858 (12.9%)** | **81/1014 (8.0%)** | **710/5872 (12.1%)** | p>0.001 |
| Intervention (Oct. 2016—Dec. 2017) | MIA + CA | 1056 | MIA rate | 37/880 (4.2%) | 9/176 (5.1%) | 46/1056 (4.4%) | p = 0.59 |
| | MIA + CA | 1056 | CA rate | 74/880 (8.4%) | 13/176 (7.4%) | 87/1056 (8.2%) | p = 0.45 |
| | MIA + CA | 1056 | Overall autopsy rate (MIA + CA) | **111/880 (12.6%)** | **22/176 (12.5%)** | **133/1056 (12.6%)** | p = 0.97 |
| Comparison of overall autopsy rate: intervention vs. non intervention | | | | p = 0.79 | p = 0.049 | p = 0.65 | |

**Table 2. Reasons why doctors did not ask for permission.**

| Motivation | Frequency |
|---|---|
| The cause of death is already known | 32/69 (46.4%) |
| The next-of-kin had already consented to an organ donation procedure | 9/69 (13%) |
| Perceived uncomfortable situation | 7/69 (10%) |
| No family present to ask permission | 6/69 (9%) |
| Doctor thought an autopsy would be too much to ask | 6/69 (9%) |

## Questionnaires

Questionnaires were distributed from September 2016 to December 2017. 505 out of 1123 (45.9%) questionnaires were returned. In the group that refused conventional autopsy or minimally invasive autopsy 413/988 (41.8%) questionnaires were returned, and in the group that gave permission 92/135 (68.1%) questionnaires were returned. In the group that gave permission for minimally invasive autopsy 40/46 (87.0%) of questionnaires were returned and in the group that gave permission for conventional autopsy 55/92 (59.8%) questionnaires were returned.

In 436/505 (86.3%) cases the doctors involved in the consent process declared that they had requested permission for autopsy. Doctors' reasons not to ask for permission are listed in Table 2, the most frequently reported reason was "the cause of death is already known" 32/69 (46.4%).

Reasons of next-of-kin for giving or denying consent for postmortem diagnostics are listed in Tables 3 and 4.

The reasons for consenting to an autopsy procedure mostly overlapped for both conventional autopsy and minimally invasive autopsy. One exception was that people consenting to minimally invasive autopsy did so more often "to contribute to research and/or contribute to medical knowledge" than those consenting to conventional autopsy (5/46 = 10.9% vs. 2/99 = 2.0%, p = 0.02).

## Discussion

To the best of our knowledge this is the only study in which acceptance of MIA and CA was prospectively investigated. The outcome is consistent with the results of a questionnaire-study by Rutty et al., which addressed the acceptance of post-mortem CT-scanning compared to invasive autopsy. [19] Although of limited relevance for our study, it nonetheless demonstrates that the public overwhelmingly preferred CT-scanning over the conventional autopsy as method for post-mortem investigation in accordance with intuitive expectation.

In this study we investigated if the introduction of a minimally invasive autopsy, consisting of CT, MRI and tissue biopsies, would increase the overall autopsy rate (combined rate for

**Table 3. Reasons of next-of-kin for giving consent.**

| Motivation | Frequency |
|---|---|
| To find out the cause of death | 65/92 (70.7%) |
| Wanting to know the severity of disease | 17/92 (18.5%) |
| Cater to treating doctor's request | 16/92 (17.4%) |
| Testing for hereditary disorders | 13/92 (14.1%) |
| Testing for presence of diseases, not related to the cause of death | 10/92 (10.7%) |
| Contribute to scientific research and/or medical knowledge | 7/92 (7.6%) |
| Other reasons | 4/92 (4.3%) |

**Table 4. Reasons of next-of-kin for denying consent*.**

| Motivation | Frequency |
|---|---|
| The cause of death is already known | 166/339 (49.0%) |
| Long illness, "the deceased has suffered enough" | 77/339 (22.7%) |
| Religious motivation | 35/339 (10.3%) |
| Autopsy is considered too invasive, scary or macabre | 35/339 (10.3%) |
| Autopsy would take too long | 8/339 (2.4%) |
| Already consented to donation procedure | 7/339 (2.1%) |
| No reason given | 24/339 (7.1%) |

* this category also contains answers from next-of-kin who gave consent for one of the autopsy methods, but nevertheless gave objections against postmortem diagnostics.

conventional autopsy and minimally invasive autopsy). We found that the introduction of minimally invasive autopsy had a small, but significant effect on the overall autopsy rate. This is important information in view of the decline in autopsy rates, which is observed worldwide since approximately 1950. In our hospital we saw a further decrease in autopsy rate of approximately 0.1% per month since the start of our study. [20]

The measured effect (2.4%) that minimally invasive autopsy had on the autopsy rate was smaller than we had expected. The reason for this is unclear: perhaps the invasiveness of the procedure isn't as important a motivation to deny consent for the next-of-kin. Alternatively, minimally invasive autopsy, using biopsies, might still be considered too invasive; or maybe any additional procedures performed on the deceased's body are considered macabre or scary. Another reason might be that low autopsy rates are attributable to a low request rate [21]. In our hospital request rates are reported to be very high, but these rates are self-reported by doctors and the real request rate might be lower than what is reported. Finally, the doctors may have requested an autopsy, but at the same time conveyed their own conviction that it was not necessary because it would not yield more information than already available. The similar percentages of doctors and next-of-kin reporting 'cause of death already know' as reason for respectively not asking for, and not consenting to post-mortem investigation, supports this contention.

An important reason why we introduced the minimally invasive autopsy in our hospital is that the population of Rotterdam, where this study was carried out, consists for a large part of non-western immigrants (around 38% in 2018). Although most religions do not outrightly prohibit autopsy, in many religious groups some inhibitions against postmortem investigations are present. [22–24] In the years when minimally invasive autopsy was not available as a stand-alone postmortem investigation we observed that the autopsy rate in people of ethnicities other than western-European was significantly lower than in people with a western-European ethnicity. When minimally invasive autopsy was available, the overall autopsy rate was the same for non- western-Europeans and 'other' ethnicities (12.6% and 12.5% respectively), and the overall autopsy rate among 'other' ethnicities was significantly higher than the autopsy rate in the years without minimally invasive autopsy (12.5% vs 8.0%, p = 0.049). This strongly suggests that minimally invasive autopsy is indeed more acceptable to people with an ethnicity other than western-European.

The main motivation for next-of-kin to withhold consent for autopsy was the assumption that the cause of death was already known, and no important questions remained to be answered. This is in line with our earlier questionnaire study. [7] It should be noted here that autopsy results differ from the presumed cause of death before autopsy in a substantial

percentage of cases. Interestingly the invasiveness of the procedure was not often mentioned as motivation against postmortem diagnostics. This might suggest that the invasiveness is not a big factor in the declining autopsy rates, contrary to what is often believed. This could be part of the reason why the postmortem consent rate of western-Europeans was not higher when minimally invasive autopsy was available.

In a study by Cox et al. the authors achieved a substantial increase in autopsy rates (from 5% to 38%). They attributed this increase in autopsy rates mainly to the study setting, which resulted in improved logistics of the consent process and a big increase in the request rate of doctors. Both factors did not play a big role in our study. We did not make major changes to the logistics of the consent process and the logistics of conventional autopsy stayed the same. The autopsy request rate in our hospital was already reported to be high before the introduction of minimally invasive autopsy. Furthermore, in some cases of the study by Cox et al. a member of the research team asked for consent. [25] It has also been suggested that pathologists should personally ask next-of-kin for permission, because they are most informed about the different procedures and can most adequately answer any questions the next-of-kin might have. [26] In our study, however, the treating physician always requested consent. We educated the doctors about the method, prior to introducing the minimally invasive autopsy hospital wide. For further questions during the study period, a researcher would be available by phone at any time. Apart from that, there was limited direct involvement of the research team in the day-to-day consent process. We think that success like that of Cox et al. and a similar increase in consent rates requires a dedicated team in each hospital with in-depth knowledge of all available postmortem diagnostic methods and all involved logistics. Members of this team could assist doctors in the consent process, or even request consent from next-of-kin in person, preferably with the treating physician present.

In studies performed in Sweden attitudes towards autopsy, organ donation and dissection (donation of the body for scientific purposes) were evaluated. Interestingly the authors found that at the time of the interview Swedes were much more positive about autopsy than they were about organ donation. [27, 28] This is in stark contrast with the current situation in the Netherlands and other western countries, where the autopsy rates are often below 10%, while registered organ donation is much more common (42% in 2018 in the Netherlands). [29] In new legislation, which will be implemented in 2020, all adults in the Netherlands are registered as an organ donor unless consent is specifically denied.

Another factor that can influence autopsy rate is the quality of the autopsy perceived by the doctors who request autopsy. It is necessary to facilitate close collaboration between clinicians and the autopsy team. This includes ensuring good communication beforehand about what can be expected of the autopsy and clear and timely information about the autopsy results afterwards. Furthermore, there must be clarity about the financial aspects of autopsies: clinicians should not have to fear that a high autopsy rate will lead to a high fee for their department. In general, an increase in autopsy rates will only be achieved if there is a positive attitude towards the autopsy among clinicians, pathologists and other involved parties. [30–32]

## Limitations

A limitation of our study was a relatively low response rate to the questionnaires. In our experience making the questionnaires obligatory for doctors to fill in after death makes the quality of the responses worse. By making it optional the doctors and families with a more positive attitude towards autopsy are more inclined to respond to the questionnaire which may lead to a bias in the answers. Another limitation was that we asked the treating physician about the motivation of the next-of-kin, because we considered it unethical to ask the bereaved family

directly after their loss. This way the motivation of the next-of-kin was investigated indirectly, through the treating clinician.

In this study all ethnicities which were not classified as western-European are classified in the category "other" ethnicities. The distinction between western-European ethnicity and all 'other' ethnicities is very broad and possibly semantically confusing. The western-European group does not include people from southern-, northern- and eastern-Europe and the USA, which are also considered as "western" in the common use of the word. Unfortunately, the group of other ethnicities was relatively small (less than 20% of the total cohort), therefore further subdividing that group resulted in low receiver operating characteristic curves for the supervised machine-learning algorithm. The Dutch surname database, used for the manual validation, contains the frequency of occurrence of surnames in 1947 and 2007. Most cases with an 'other' ethnicity were migrant workers who moved to the Netherlands after the second world war. This corresponds with the data from the Personal Records Database from the municipality of Rotterdam: over 50% of the population has a migration background, and within this group roughly 75% has a non-western background.

Information about the minimally invasive autopsy was distributed at the start of the cohort. Doctors in our hospital were already familiar with the procedure, because it had already been validated in the years prior to this cohort. Nevertheless, in a hospital environment with constant changes in personnel, the familiarity with minimally invasive autopsy and the ins-and-outs of this new procedure were suboptimal. We feel that, for a reliable measurement of the effect of minimally invasive autopsy on the overall autopsy rate, a longer period of inclusion is necessary, so that doctors and the public become more familiar with the procedure. In this light, the results from our current cohort should be seen as a baseline measurement for our hospital and comparable hospitals.

## Recommendations

We recommend that MIA should be offered in populations with a high proportion having a non-Western background, and in countries where the autopsy meets a great deal of objection, like in Islamic countries. We also recommend that MIA is carried out in specialized centers because of the expertise it requires on the part of radiologists and pathologists to take and interpret the image-guided biopsies, and the costliness of equipment (CT and MRI).

## Conclusion

In this study we investigated if the introduction of a minimally invasive autopsy consisting of CT, MRI and biopsies would lead to an increase in the overall autopsy rate (conventional autopsy and minimally invasive autopsy). We found that the introduction of minimally invasive autopsy had a small, but significant effect on the autopsy rate. Furthermore, the minimally invasive autopsy appears to be more acceptable than conventional autopsy among people with an ethnicity other than western-European.

## Author Contributions

**Conceptualization:** I. M. Wagensveld, A. C. Weustink, B. M. Blokker, M. G. M. Hunink, J. W. Oosterhuis.

**Formal analysis:** I. M. Wagensveld, J. A. Kors.

**Funding acquisition:** J. W. Oosterhuis.

**Investigation:** I. M. Wagensveld, A. C. Weustink.

**Methodology:** I. M. Wagensveld, J. A. Kors, B. M. Blokker, M. G. M. Hunink.

**Project administration:** J. W. Oosterhuis.

**Supervision:** A. C. Weustink, M. G. M. Hunink, J. W. Oosterhuis.

**Writing – original draft:** I. M. Wagensveld.

**Writing – review & editing:** I. M. Wagensveld, A. C. Weustink, J. A. Kors, B. M. Blokker, M. G. M. Hunink, J. W. Oosterhuis.

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
