## [Decision Letter · Decision Letter 0]

11 Nov 2019

PONE-D-19-23443

Effect of minimally invasive autopsy and ethnic background on consent rate for postmortem investigation in adult deceased patients: a prospective single center before-after study

PLOS ONE

Dear Mr. Wagensveld,

Thank you for submitting your manuscript to PLOS ONE. After careful consideration, we feel that it has merit but does not fully meet PLOS ONE’s publication criteria as it currently stands. Therefore, we invite you to submit a revised version of the manuscript that addresses the points raised during the review process.

The manuscript has been assessed by two reviewers; their comments are available below.

The reviewers are positive about the study but have requested some additions and clarifications, including additional information on the procedures undertaken and further discussion of the policy implications of the findings.

In addition to the comments raised by the reviewers, please provide further information under the Methods section on the questionnaires employed as part of the study, please indicate whether previously available questionnaires were employed, or the questionnaires were developed for this study, and if the latter, indicate whether the questionnaires were validated and how.

Could you please carefully revise the manuscript to address the concerns raised?

We would appreciate receiving your revised manuscript by Dec 24 2019 11:59PM. Please include the following items when submitting your revised manuscript:

We look forward to receiving your revised manuscript.

Kind regards,

Iratxe Puebla

Senior Managing Editor, PLOS ONE

Journal Requirements:

2. Please provide additional details regarding participant consent.

In the ethics statement in the Methods and online submission information, please ensure that you have specified (a) whether consent was informed and (b) what type you obtained (for instance, written or verbal, and if verbal, how it was documented and witnessed).

If your study included minors, state whether you obtained consent from parents or guardians.

If the need for consent was waived by the ethics committee, please include this information.

Reviewers' comments:

Reviewer's Responses to Questions

**Comments to the Author**

1. Is the manuscript technically sound, and do the data support the conclusions?

Reviewer #1: Yes

Reviewer #2: Yes

2. Has the statistical analysis been performed appropriately and rigorously? 

Reviewer #1: I Don't Know

Reviewer #2: Yes

3. Have the authors made all data underlying the findings in their manuscript fully available?

Reviewer #1: Yes

Reviewer #2: No

4. Is the manuscript presented in an intelligible fashion and written in standard English?

Reviewer #1: Yes

Reviewer #2: Yes

5. Review Comments to the Author

Reviewer #1: 1. I think there is a need to revise the title. Currently, the title is little long.

2. In objectives (abstract section), you have not highlighted that you are also measuring the consent rates among different ethnicity. You might want to add that..

3. In introduction section line no. 56, you have used the phrase" determining acceptance rates'; you can think of using this in your title. Whatever you choose to write; you should remain consistent.

4. In introduction section; line no 60-62; you have added one more objective. This is not stated in your title and abstract. You might want to add it there too

5.I think the description of MIA and conventional autopsy can be moved to introduction section from the methods section

6.I think in methods section, you should have the heading of primary and secondary outcomes; instead of acceptance and questionnaires

7. Also, I think you should talk about the procedures in detail. How this was done. Method section needs more clarity.

8. I believe the discussion section needs some more work; in terms of comparing and contrasting similar studies that have been conducted on this domain.

9. It would be good if you can highlight some policy implications for this study.

10. I have not commented on the results section and stats

Reviewer #2: It is a well written manuscript with sound statistical analysis. Just one suggestion to use word "Acceptance" in title of the paper

6. PLOS authors have the option to publish the peer review history of their article (what does this mean?). If published, this will include your full peer review and any attached files.

Reviewer #1: Yes: Anam Feroz

Reviewer #2: No

---

## [Author Response · Author response to Decision Letter 0]

1 Feb 2020

Reviewer #1: 

1. I think there is a need to revise the title. Currently, the title is little long.

Answer: 

We changed the title to: ‘Effect of minimally invasive autopsy and ethnic background on acceptance of clinical postmortem investigation in adults’

2. In objectives (abstract section), you have not highlighted that you are also measuring the consent rates among different ethnicity. You might want to add that..

Answer: 

Added this to the objectives section in the abstract.

3. In introduction section line no. 56, you have used the phrase" determining acceptance rates'; you can think of using this in your title. Whatever you choose to write; you should remain consistent.

Answer:

We use the word ‘acceptance’ in the new title. Where possible we changed the wording in the text to ‘acceptance’ instead of consent rate, however, the word acceptance has a broader meaning than just consent, therefore we use the word ‘consent’ for the situation of giving specific consent for the procedure and the word ‘acceptance’ for the values and opinions of next-of-kin with regards to postmortem diagnostics that lead to consenting to postmortem investigation. 

Since in our study postmortem investigation was always carried out when consent was given, the terms ‘consent rate’ and ‘autopsy rate’ are interchangeable in terms of numbers. Therefore, we now consistently use the term ‘autopsy rate’ instead of ‘consent rate’ (where appropriate), in our methods, results and discussion section, because this term is most commonly used in the literature. Whenever we use the term ‘overall autopsy rate’ it includes conventional- and minimally invasive autopsy.

4. In introduction section; line no 60-62; you have added one more objective. This is not stated in your title and abstract. You might want to add it there too

Answer:

We added the objective to measure consent rate among different ethnicities in the objectives in the abstract. The questionnaires were only carried out as a secondary analysis and therefore we did not add those results to the abstract, to limit the amount of words in the abstract and to focus on the main objectives and results.

5.I think the description of MIA and conventional autopsy can be moved to introduction section from the methods section.

Result: 

We feel that the description of the conventional autopsy and minimally invasive autopsy belongs in the methods section. A very concise description of the minimally invasive autopsy is already present in the introduction section.

6.I think in methods section, you should have the heading of primary and secondary outcomes; instead of acceptance and questionnaires

Answer:

We changed the headings in the methods section according to the suggestions of the reviewer.

7. Also, I think you should talk about the procedures in detail. How this was done. Method section needs more clarity.

Answer: 

We made the choice to keep the procedure details about the different methods of the conventional autopsy and minimally invasive autopsy concise, because these are not relevant to the current study. In response to the reviewer, we added a line in the methods section that explains the main difference between the minimally invasive autopsy and the conventional autopsy. Any additional information can be found in the quoted previously published articles (among them publications in Plos One) if the reader is interested, but these details are not necessary for understanding our study or replicating the results.

8. I believe the discussion section needs some more work; in terms of comparing and contrasting similar studies that have been conducted on this domain.

Anwer: To the best of our knowledge this is the only study in which acceptance of MIA and CA was prospectively investigated. The outcome is consistent with the results of a questionnaire-study by Rutty et al. (J Forensic Legal Med 2011), which addressed the acceptance of post-mortem CT-scanning compared to invasive autopsy. Although only partially relevant for our study, it nonetheless demonstrates that the public overwhelmingly preferred CT-scanning over the conventional autopsy as method for post-mortem investigation in accordance with intuition. We added a paragraph at the start of the discussion about the comparison with the available literature. 

9. It would be good if you can highlight some policy implications for this study.

Answer: There is consensus that autopsies are important for reliable health statistics for health care policymaking. In populations with a substantial proportion having a non-Western background the data will be biased towards the Western population to the disadvantage of the non-Western part. A way of addressing this problem is to apply autopsy techniques that are acceptable to both groups. Our study shows that with the MIA applied here the autopsy rates among Western and non-Western people are similar. We recommend that MIA should be offered in populations with a high proportion having a non-Western background, and in countries where the autopsy meets a great deal of objection, like in Islamic countries. We also recommend that MIA is carried out in specialized centers because of the expertise it requires on the part of radiologists and pathologists to take and interpret the image-guided biopsies, and the costliness of equipment (CT and MRI). We added a paragraph at the end of our discussion section with policy implications. 

10. I have not commented on the results section and stats

Reviewer #2: It is a well written manuscript with sound statistical analysis. Just one suggestion to use word "Acceptance" in title of the paper

Answer: 

We changed the title to: 

‘Effect of minimally invasive autopsy and ethnic background on acceptance of clinical postmortem investigation in adults’

---

## [Editor Report · Decision Letter 1]

27 Apr 2020

Effect of minimally invasive autopsy and ethnic background on acceptance of clinical postmortem investigation in adults

PONE-D-19-23443R1

Dear Dr. Wagensveld,

We are pleased to inform you that your manuscript has been judged scientifically suitable for publication and will be formally accepted for publication once it complies with all outstanding technical requirements.

With kind regards,

Mateusz K. Holda, MD, PhD, DSc

Academic Editor

PLOS ONE

---

## [Editor Report · Acceptance letter]

29 Apr 2020

PONE-D-19-23443R1 

Effect of minimally invasive autopsy and ethnic background on acceptance of clinical postmortem investigation in adults 

Dear Dr. Wagensveld:

I am pleased to inform you that your manuscript has been deemed suitable for publication in PLOS ONE. Congratulations! Your manuscript is now with our production department. 

With kind regards,

on behalf of

Dr. Mateusz K. Holda 

Academic Editor

PLOS ONE